# Effects of Trehalose-6-Phosphate Synthase on the Reproduction and Development of *Nilaparvata lugens* and Its Molecular Mechanism

**DOI:** 10.3390/insects16121195

**Published:** 2025-11-24

**Authors:** Ye Han, Fan Zhong, Xinyu Zhang, Yuya Zhang, Yanfei Zhou, Liwen Guan, Yongkang Liu, Yi Zhang, Xinyi Zhang, Min Zhou, Bin Tang

**Affiliations:** College of Life and Environmental Sciences, Hangzhou Normal University, Hangzhou 311121, China; 19011285720@163.com (Y.H.); zf605261118@163.com (F.Z.);

**Keywords:** RNAi, *Nilaparvata lugens*, trehalose-6-phosphate synthase, reproduction, signal pathway, lipid metabolism

## Abstract

*Nilaparvata lugens* (Brown rice planthopper, BPH), a migratory rice pest with strong fecundity and insecticide resistance, severely threatens global rice production. This study aimed to explore how trehalose-6-phosphate synthase—a key gene in trehalose biosynthesis—affects *N. lugens*’s reproduction and related molecular mechanisms. We used RNA interference (RNAi) to silence three trehalose-6-phosphate synthase paralogs (*TPS1*, *TPS2*, *TPS3*) in female *N. lugens*. Results showed that *TPS* silencing significantly delayed ovarian development, prolonged the pre-oviposition period, reduced total egg production and offspring hatch rate, and downregulated the expression of vitellogenin (Vg) and its receptor (VgR)—critical for oocyte maturation. It also disrupted hormonal signaling (juvenile hormone and 20-hydroxyecdysone pathways) and nutrient-sensing cascades (insulin/IGF and TOR pathways), as well as altered lipid metabolism. This dual effect of impairing energy supply and hormonal regulation highlights *TPS* gene as a promising molecular target for developing eco-friendly strategies to control *N. lugens*, which is vital for safeguarding rice yield and food security.

## 1. Introduction

*Oryza sativa* (rice) underpins the food security and livelihoods of hundreds of millions worldwide, especially in impoverished regions. However, rice production is continually imperiled by a range of insect pests and diseases that erode both yield and grain quality [1,2]. Chief among these threats is *Nilaparvata lugens* (Brown rice planthopper, BPH), the most devastating migratory pest of rice, closely followed by its congener, *Sogatella furcifera* (White-backed planthopper, WBPH), a potent vector of microbial and viral pathogens. Together, they inflict staggering yield losses across rice-growing regions [3,4]. Specifically, adults and nymphs of *N. lugens* congregate at the rice root zone, inserting their piercing–sucking stylets into phloem vessels to draw sap at rates that can exceed three times their own body weight per day [5,6]. Concurrently, salivary phenolic toxins breach cellular membranes, inflicting cascading physiological injury throughout the plant [7]. Furthermore, *N. lugens* excretes copious honeydew that becomes a fertile substrate for saprophytic fungi; the resulting sooty film blankets leaf surfaces and sharply curtails rice photosynthetic efficiency [8]. In addition, *N. lugens* exhibits remarkable fecundity: during oviposition, female adults repeatedly pierce rice stems, generating wounds that accelerate water loss in the host plant and provide entry points for destructive pathogens, including sheath blight and rice blast pathogens [9]. In recent years, the pest has devastated millions of hectares of paddies and acquired resistance to multiple commercial insecticides [10]. Consequently, the development of next-generation, biologically based green pesticides has become an urgent imperative.

RNA interference (RNAi) has emerged as an indispensable tool for functional genomics and a transformative platform for species-specific pest management [11]. Microinjection remains the gold-standard method for double-stranded RNA (dsRNA), enabling direct introduction of these molecules into embryos or the hemocoel. This allows hemolymph circulation to rapidly disseminate the trigger throughout the insect body [12,13,14], and consistent with this, extensive studies have demonstrated that silencing the *TPS* and *TRE* genes disrupts chitin metabolism in *Acyrthosiphon pisum* [15], while suppression of *HpNAG* and *HpNAGK* reduces cuticle thickness and chitin content in the coleopteran *Holotrichia parallela Motschulsky* [16]. Similarly, silencing the *Ago1* and *Ago2* genes in *S. furcifera* disrupts ecdysis and leads to lethality [17]. Thus, suppressing the expression of key genes can prove lethal to insects, establishing RNAi as a promising, gene-based weapon for next-generation pest control.

Trehalose functions as the principal hemolymph sugar and primary energy reserve in insects, orchestrating growth during chronic cold stress and serving as a central regulator of development, stress tolerance, flight capacity, and reproductive output [18,19,20]. Trehalose biosynthesis hinges on the trehalose-6-phosphate synthase gene (*TPS*). In adults, three *TPS* paralogs are selectively expressed in the head, legs, wings, and epidermis [21]. Silencing *TPS* expression arrests trehalose formation and drives a compensatory rise in free glucose [20]. The expression level of the *TPS* gene directly governs trehalose concentration; when insects face starvation or high energy expenditure, *TPS* activity is promptly up-regulated, accelerating trehalose synthesis to maintain hemolymph glucose homeostasis [22,23]. During overwintering, *Helicoverpa armigera* markedly up-regulates *TPS* expression and accumulates trehalose, a metabolic safeguard that underpins *H. armigera*’s cold-hardiness [24,25]. Additionally, starvation triggers a marked rise in hemolymph trehalose concentration in *Spodoptera exigua* [26]. Similarly, during the pupal stage, *Bombyx mori* accumulates trehalose to fuel adult eclosion; silencing *TPS* disrupts ecdysone signaling and arrests pupation altogether [27]. In contrast, feeding decisions in *A. pisum* are dictated by the dynamic balance of intracellular trehalose and glucose levels [21]. Collectively, these examples all illustrate the importance of trehalose for insects. Silencing *TPS* genes in insects, exemplified by *Heortia vitessoides* and *Tribolium castaneum*, precipitates lethality and severe malformations, underscoring the gene’s pivotal role in coordinating energy and chitin metabolism [28,29]. Inhibiting trehalase reduces the synthesis of trehalose. Trehalose can be hydrolyzed into glucose, which provides energy for these physiological activities of insects, thereby ensuring the normal progress of the reproductive process and playing a crucial role in reproduction [30,31].

Behavioral assays show that *N. lugens* and *S. furcifera* exhibit a strong preference for rice seedlings as oviposition sites. However, as the crop matures, the oviposition propensity of *S. furcifera* decreases precipitously, while *N. lugens* maintains continuous egg-laying activity [32]. Vitellogenin (Vg) and its receptor (VgR) are key regulators of insect reproduction, as they control oocyte maturation, embryogenesis, and ultimately mediate population dynamics [33,34]. Studies in *Periplaneta americana* have shown that during yolk deposition, juvenile hormone (JH) and 20-hydroxyecdysone (20E) operate downstream of indoleamine- and peptide-mediated cues to orchestrate vitellogenesis [35]. More broadly, insect vitellogenesis is regulated by an integrated network of hormonal and nutrient-sensing pathways, including JH, 20E, the insulin/insulin-like growth factor signaling (IIS) pathway, and the target of rapamycin (TOR) pathway. These pathways collectively govern yolk synthesis and deposition [36,37].

This study had two interrelated objectives. First, to clarify the role of the *TPS* gene in *N. lugens* reproduction, we used RNAi to silence the *TPS* gene, then quantified changes in the insect’s reproductive output and delineated transcriptional reprogramming of key reproductive signaling pathways. Second, to propose a TPS-targeted control strategy for this pest: by disrupting *N. lugens*’ trehalose metabolism via *TPS* silencing, we aimed to validate the *TPS* gene as a viable molecular target and lay the foundation for developing a potent, species-specific control measure against this devastating agricultural pest.

## 2. Materials and Methods

### 2.1. Source and Breeding of N. lugens

Rice cultivar Taichung Native 1 (TN1) served as the host plant *N. lugens* was originally field-collected from paddies at the China National Rice Research Institute and maintained on TN1 seedlings for more than 60 consecutive generations under controlled conditions. TN1 seeds were surface-sterilized and imbibed in sterile water at warm water for 24–48 h. During the warm season, germinated seeds were grown on autoclaved soil for a week; in the cool season, perform hydroponic cultivation in an artificial climate chamber. Virgin males and females were paired in mesh cages containing fresh TN1 seedlings. After mating, females oviposited into the leaf sheaths, and eggs hatched 6–7 days later. The environmental conditions of the artificial climate chamber are set as follows: temperature 27 °C ± 1 °C, relative humidity 65 ± 5%, and photoperiod 18 L:6 D (Light:Dark). Replace fresh rice seedlings in a timely manner during the breeding process to ensure sufficient food supply for the population.

### 2.2. Total RNA Isolation and cDNA Synthesis

Total RNA was extracted from adult *N. lugens* using Trizol reagent [38]. Each treatment consisted of three biological replicates, each comprising five adults. The purity and concentration of RNA were detected by Nanodrop 2000 (Thermo Fisher, Waltham, MA, USA), and the integrity of RNA was detected by 1% agarose gel electrophoresis. If the RNA quality is qualified, it will be stored at −80 °C for subsequent experiments. The final concentration of each RNA was adjusted to 1000 ng/μL for subsequent cDNA synthesis. RNA was used as template, and the prime script RT reagent Kit with gDNA Eraser Kit (Takara, Japan) was used for cDNA transcription reaction.

### 2.3. Synthesis and Purification of dsRNA

The specific primers for *TPS1*, *TPS2* and *TPS3* of *N. lugens* were designed using Primer Premier 5 software (Premier Biosoft, Canada) in Table 1, in which the T7 promoter sequence was GGATCCTAATACGACTCACTATAGG. Double-stranded RNA was synthesized according to the method of T7 Ribo MAXTM Express RNAi System Kit (Promega, America). The same method was used to synthesize the dsRNA of green fluorescent protein gene (GFP) as the control group.

### 2.4. Microinjection and Post-Injection Rearing of N. lugens

Prepare a capillary glass tube with an inner diameter of 0.5 mm, and use a P-1000 needle puller (Sutter instrument, America) to pull the glass tube into a micro injection needle with a thin tip. Newly emerged *N. lugens* females were briefly anesthetized with CO_2_ and microinjected under a stereomicroscope (Leica, Germany) using a calibrated microinjector. Injections were delivered into the soft membranous region between the meso- and metathoracic coxae of freshly anesthetized *N. lugens* females. A fine-tipped glass needle (tapered with forceps to prevent bending and minimize mechanical damage) was used for each microinjection. Each experiment was performed with three replicates, and five adult *N. lugens* were injected for each replicate. Two injection treatments were established: (i) dsGFP (negative control) and (ii) an equimolar cocktail (1:1:1) of the three synthesized dsRNAs—dsTPS1, dsTPS2, and dsTPS3—designated dsTPSs [39,40].

### 2.5. Quantitative Real-Time Polymerase Chain Reaction (qRT-PCR)

Gene-specific qRT-PCR primers were designed with Primer Premier 5, and the *Actin* gene served as the endogenous reference. Primer sequences are listed in Table 2 (all primer efficiencies are greater than 90%). The qRT-PCR reaction system was as follows: TB Green Premix Ex Taq 5 μL, ddH_2_O 3.6 μL, forward primer F 0.2 μL, reverse primer R 0.2 μL, cDNA 1 μL. The qRT-PCR reaction was performed on a Bio-Rad CFX system using the following thermal profile: pre-denaturation at 95 °C for 30 s, denaturation at 95 °C for 5 s, extension at 60 °C for 30 s, and 35 cycles. Amplicon specificity was verified via amplification and melt-curve analyses, and relative gene expression was quantified by the 2^−ΔΔCT^ method [41].

### 2.6. Integrated Ovarian Transcriptomic and Metabolomic Profiling

Transcriptome sampling: Females microinjected with dsRNA and untreated males were paired 1:1 on fresh rice seedlings; 3 days later, ovaries were dissected under a stereomicroscope. Three biological replicates were prepared, each pooling 5–10 ovaries. Total RNA was isolated as described in Section 2.2, then the RNA sample was sent to the Beijing Genomics Institute (Shenzhen, China) for transcriptome sequencing (sequencing platform: DNBSEQ, sequencing depth ≥ 60×).

Metabolome sampling: Microinjected females and untreated males were paired 1:1 on fresh rice seedlings. After 3 days, ovaries were dissected under a Leica EZ4 HD stereomicroscope, yielding six biological replicates of 15–25 ovaries each. The ovaries were frozen in liquid nitrogen, and the samples were sent to the Beijing Genomics Institute (Shenzhen, China) for non-targeted metabolomics sequencing.

Single-stranded circular DNA library (a transcriptome library constructed from mRNA), using a polyA enrichment strategy which binds to the polyA tail of mRNA via OligodT magnetic beads to enrich mRNA with a polyA structure and exclude rRNA interference.

Total RNA-seq raw reads were first filtered using SOAPnuke software (BGI Shenzhen) to remove reads with low quality, adapter contamination, and excessively high content of unknown base N, resulting in clean reads. Differentially expressed genes (DEGs) were analyzed using DESeq2 software with the parameter setting of q-value (adjusted *p*-value) ≤ 0.05 [42]. The screened DEGs were subjected to KEGG (Kyoto Encyclopedia of Genes and Genomes) pathway analysis. In addition, this experiment employed Principal Component Analysis (PCA) to examine the overall distribution of individual samples within each group and the degree of dispersion between groups.

The following experimental steps were completed by the company: Using a liquid extracted from a mixture of methanol, ethanol and water, extract a sample of brown-flush ovaries, add two sterilized small steel beads to grind, and then grind the grinding liquid in an ultrasound bath at 4 °C. Mix the methanol and water according to the volume ratio of 9:1, and prepare a solvent; take the grinding liquid, put it in a frozen vacuum enrichment, then add 200 μL of the solvent to dissolve, then concentrate for 15 min, take the liquid to transfer to the above sample bottle. In addition, 20 μL of liquid is extracted from each sample separately and mixed together to form a QC sample, and LC-MS analysis is carried out in sync with each treatment group sample to assess the repeatability and stability of the analysis process. The extracted metabolites were separated and tested using the Q Exactive HF High Resolution Spectrometer (Thermo Fisher Scientific, Waltham, MA, USA) in both positive and negative ions.

### 2.7. Detection of the Fecundity of N. lugens

Ovarian dissection and imaging: Injected females were paired 1:1 with untreated males. On days 3, 5 and 7 post-mating, females were briefly anesthetized with CO_2_ and ovaries were dissected intact under a stereomicroscope (Leica, Germany).

Female *N. lugens* after injection were paired and reared with untreated male *N. lugens* at a 1:1 ratio. The oviposition status was observed daily, and the number of days from pairing to the start of oviposition is defined as the pre-oviposition period. On day 3 post-mating, ovaries were dissected, staged, and the number of mature oocytes was quantified [43]; more than 15 females were examined per treatment. Based on oocyte maturity, color, and number within the ovarioles, the adult ovaries were classified into five distinct stages: milky transparent (Grade I), vitellogenic (Grade II), mature (Grade III), egg-laying (Grade IV), and late egg-laying (Grade V).

Offspring hatch-rate assessment: Injected females were paired 1:1 with untreated males. Three days later, each mated female was transferred to a feeding device containing fresh TN1 seedlings immersed in clean water, at a density of 1–3 females per device. Hatch rates of the resulting eggs were recorded. Each trial comprised ≥ 9 replicates. Collect the eggs laid by the female of *N. lugens* from the 3rd day to the 6th day after eclosion, and then remove the adults. Count the egg hatching of *N. lugens* every 24 h under the condition of artificial climate chamber until no further nymphs emerge. Unhatched eggs were examined and categorized under a stereomicroscope.

### 2.8. Determination of Triglyceride Content

Injected females were paired 1:1 with untreated males. After 72 h, fat bodies and ovaries were dissected. Each treatment comprised three biological replicates, each pooling 5–8 tissues (fat bodies or ovaries) from *N. lugens*.

Determination of triglycerides in fat body: Triglyceride (TG) content in fat bodies was quantified using Triglyceride assay kit (Nanjing Jiancheng, China). The quantity of quinones generated in the reaction correlates directly with triglyceride content, with higher levels yielding darker colorimetric signals. The calculation formula of triglyceride content is as follows:Triglyceride content (mmol/L) = (Sample OD value − Blank OD value)/(Calibration OD value blank OD value) × Concentration of calibrator (mmol/L)

Triglyceride (TG) quantification in ovaries: The TG assay procedure was identical to that described above. However, given that ovaries are not high-fat tissues, protein content was also measured in the pre-treated samples to normalize TG levels. Protein concentration was determined using a BCA Protein Assay Kit (Beyotime, Shanghai, China). Absorbance was measured at 562 nm using a microplate reader. TG content was then calculated based on the protein concentration of the sample as follows:Triglyceride content (mmol/L) = [(Sample OD value − Blank OD value)/(Calibration OD value blank OD value) × Concentration of calibrator (mmol/L)]/Protein concentration of the sample (gprot/L)

### 2.9. Data Analysis

Statistical significance of the data was assessed using IBM SPSS Statistics 20 software, with normality and homogeneity of variance evaluated. Differences between control and treatment groups were compared using one-way analysis of variance (ANOVA) or independent sample *t*-test. Post hoc tests were performed using the Tukey method for one-way ANOVA, with different letters indicating significant differences between groups (*p* < 0.05). For the independent sample *t*-test, “*” denoted a significant difference when *p* < 0.05, “**” indicated an extremely significant difference when *p* < 0.01, “***” represented an even more extremely significant difference when *p* < 0.001, and “ns” indicated no significant difference. Data are presented as mean ± SD. GraphPad Prism version 9.0 software was used for data visualization (all data conformed to a normal distribution).

## 3. Results

### 3.1. RNAi Efficacy and Transcriptomic-Metabolomic Analysis

On day 3 post-injection, females were sampled for RNAi validation. Quantitative real-time PCR (qRT-PCR) revealed significant downregulation of the three trehalose-6-phosphate synthase genes—*TPS1*, *TPS2*, and *TPS3*—indicating effective dsRNA-mediated silencing (Figure 1A). Through the analysis of ovarian transcriptome and metabolome, we found that the correlation coefficients within the dsGFP and dsTPSs groups exceeded 99% and 96%, respectively, confirming robust sample reproducibility across both treatments (Figure 1B). Moreover, analysis of inter-group transcriptomic correlation showed that the correlation coefficient between the dsGFP and dsTPSs groups exceeded 94%, confirming high transcriptomic similarity between the two groups. Consistent with this result, principal component analysis (PCA) demonstrated that all samples from the two treatment groups were tightly clustered in the PCA space (Figure 1C), which further verified the high degree of similarity in their transcriptomic profiles. The results of KEGG analysis are pivotal for elucidating the biological pathways and molecular mechanisms associated with gene functions. The results indicated significant enrichment in pathways related to Fatty acid elongation, Fatty acid metabolism, Proximal tubule bicarbonate reabsorption, and Fatty acid degradation (Figure 1D). Integrating KEGG pathway enrichment analysis from both positive and negative ion modes, the differentially abundant metabolites between the dsTPSs and dsGFP groups were primarily enriched in pathways related to the biosynthesis of Glutathione metabolism, Ascorbate and aldehyde metabolism, etc. (Figure 1E,F). In summary, the transcriptomic and metabolomic data are highly consistent, providing crucial insights into the role of glucose metabolism in regulating the fecundity of *N. lugens* (The transcriptomic-metabolomic analysis and metabolomics run metrics are provided in the Appendix A).

### 3.2. Impact of Trehalose-6-Phosphate Synthase Gene on Ovarian Development in Female N. lugens

Examining the ovarian morphology of *N. lugens*, we observed that compared to the control group injected with dsGFP, the ovarian area of *N. lugens* injected with dsTPSs was significantly reduced on days 3 and 5. Additionally, the number of mature eggs in development was lower, and ovarian development was insufficient. However, by day 7, ovarian development appeared sufficient (Figure 2A). On day 3 post-injection, ovaries were dissected from females injected with dsGFP, staged, and mature oocyte counts were recorded. Most ovaries from dsGFP-injected females reached Grade III, with only a few remaining at Grade II. In contrast, ovaries from females injected with dsTPSs exhibited a significant increase in Grade I and II (Figure 2B) and a marked reduction in mature eggs (Figure 2C), indicating that dsTPSs treatment impeded ovarian growth and significantly inhibited early-stage ovarian development in *N. lugens*.

### 3.3. Impact of Trehalose-6-Phosphate Synthase Gene Silencing on Egg Production and Hatch Rate in Female N. lugens

Following injection with dsGFP and dsTPSs, the mean pre-oviposition periods of *N. lugens* were 2.65 d and 3.53 d, respectively. The pre-oviposition period of the dsTPSs-injected group was significantly longer than that of the dsGFP control group (Figure 3A). Figure 2A shows that the ovarian development of brown planthoppers injected with dsTPSs is inhibited. Correspondingly, the results showed that compared with the control group injected with dsGFP, the total egg production of *N. lugens* injected with dsTPSs was significantly reduced (Figure 3B). In addition, the hatching rate of *N. lugens*’ offspring was significantly reduced in the treatment group injected with dsTPSs (Figure 3C). It can be seen from the figure that the hatching failure of *N. lugens* may be related to the development error of the anterior–posterior axis of the egg or the eggshell (Figure 3D). Three days after the dsTPSs injection, the relative expression levels of *Vg* and *VgR* were significantly downregulated (Figure 3E,F), which was associated with the observed insufficient development of egg granules (Figure 3D) and ovarian hypoplasia (Figure 2A).

### 3.4. Impact of Trehalose-6-Phosphate Synthase Gene on JH and 20E Signaling Pathways

Three days post-injection of dsTPSs, the juvenile hormone biosynthetic gene *JHAMT* showed significantly downregulated expression compared to controls (0.01 < *p* ≤ 0.05; Figure 4A), while the JH intracellular receptor-encoding gene *Met* was profoundly reduced (*p* ≤ 0.001; Figure 4B). Additionally, the 20-hydroxyecdysone receptor genes *ECR* and *USP* were both downregulated (*p* ≤ 0.001; Figure 4C,D). These findings indicate that dsTPSs injection attenuates both JH and 20E signaling in *N. lugens*.

### 3.5. Impact of Trehalose-6-Phosphate Synthase Gene on Nutrient Signaling Pathways

On the third day post-dsTPSs injection, the expression level of the insulin receptor gene *InR1* in *N. lugens* exhibited a highly significant downward trend compared to the control group (0.001 < *p* ≤ 0.01) (Figure 5A). Similarly, the expression level of another insulin receptor gene, *InR2*, also decreased significantly (0.01 < *p* ≤ 0.05) (Figure 5B). As a core molecule in the regulatory pathway of cell growth and metabolism, the mRNA level of the *TOR* gene was significantly reduced compared to the control group (0.01 < *p* ≤ 0.05) (Figure 5C). *S6K*, a downstream effector protein of mTOR, showed no significant difference in gene expression compared to the control group (*p* > 0.05) (Figure 5D).

### 3.6. Impact of Trehalose-6-Phosphate Synthase Gene on Lipid Metabolism

The experimental results indicated that, on day 3 post-dsTPSs injection, triglyceride (TG) content in fat bodies did not significantly differ from controls (Figure 6A). However, TG levels in ovaries were significantly elevated (Figure 6B). The impact of *TPS* silencing on lipid metabolism in *N. lugens* was further examined via qRT-PCR. Results indicated that fatty acid synthase gene (*Fas*) mRNA levels were significantly reduced (0.01 < *p* ≤ 0.05) on day 3 post-dsTPSs injection (Figure 6C). Conversely, adipokinetic hormone gene (*AKH*), which regulates lipid mobilization, exhibited a downward trend but no significant difference compared to controls (Figure 6D).

## 4. Discussion

Characterized by its migratory behavior, prolific reproductive capacity, propensity for outbreaks, and robust resistance to insecticides, *N. lugens* poses a significant threat to rice production. Effective management of this pest must adhere to the principle of “prevention first, integrated control”. In recent years, research has increasingly focused on the development of novel, eco-friendly pesticides [30,44,45,46]. This study, commencing with the reproductive biology of *N. lugens*, employs RNAi technology to silence the *TPS* gene, thereby disrupting trehalose synthesis in vivo. This intervention results in diminished reproductive capacity in adult insects, laying a foundation for the potential development of RNAi-based biopesticides in the future.

The experimental results indicated that the differentially abundant metabolites between the dsTPSs and dsGFP groups were primarily enriched in metabolic pathways, including glutathione metabolism and ascorbate/aldarate metabolism (Figure 1E,F). Glutathione (GSH), an essential tripeptide, is a cornerstone in enzymatic detoxification, crucial for modulating reactive oxygen species (ROS) levels and implicated in myriad key physiological functions [47,48,49]. Glutathione is indispensable for embryonic development and modulates reproductive efficiency by regulating oxidative stress, energy metabolism, and signal transduction [50,51,52,53,54]. Similarly, ascorbic acid has long been linked to fertility: its interplay with aldose acid metabolism fine-tunes germ cell quality, hormonal balance, and embryonic development, thereby influencing reproductive outcomes [55]. During oocyte maturation, these metabolites further support reproductive processes by shielding oocytes from ROS-mediated damage and ensuring adequate nutrient supply via their antioxidant properties [47,56,57]. In the present study, KEGG analysis revealed that differentially expressed genes (DEGs) in the dsTPSs group were significantly enriched in pathways related to Fatty acid metabolism and Fatty acid degradation (Figure 1D), consistent with the role of *TPS* genes in metabolic regulation. To validate *TPS* gene silencing efficiency, we detected the relative expression of three trehalose-6-phosphate synthase genes (*TPS1*, *TPS2*, and *TPS3*) in *N. lugens* on day 3 post-RNAi: all three genes showed significant downregulation (Figure 1A). This result differs from a related study where *TPS1* expression in *N. lugens* increased on day 3 following dsTPSs injection [45]; we propose that this discrepancy may stem from differences in the injected dsTPSs concentrations, a factor known to affect RNAi efficiency. We further investigated key genes and metabolites in fatty acid metabolism. *Fas*, a highly conserved key gene in fatty acid biosynthesis that promotes lipid accumulation during the lipid storage phase and maintains lipid homeostasis by upregulating lipid metabolism [58,59], showed significant downregulation in the dsTPSs group (Figure 6C). Consistent with this, triglyceride (TG) levels— a core indicator of lipid metabolism in insects [60]—exhibited tissue-specific changes: fat body TG content showed no significant difference (Figure 6A), while ovarian TG levels increased markedly (Figure 6B). We hypothesize that this elevation in ovarian TG may reflect aberrant lipid metabolism, a phenomenon often associated with reproductive dysfunction. Notably, triglycerides stored in the fat body are progressively hydrolyzed into ATP by lipase [61,62]. Combined with our findings of *TPS* gene silencing, *Fas* downregulation, and altered TG distribution, these results collectively suggest that silencing *TPS* genes disrupts fatty acid metabolism pathways. This disruption may reduce ATP production, thereby modulating the reproductive processes of *N. lugens*.

Trehalose, known as the “blood glucose” of insect hemolymph, acts as a central energy source for ovarian development and a key protector against oocyte stress [31]. Its synthesis in insect ovaries follows a well-defined pathway: trehalose-6-phosphate synthase catalyzes the formation of trehalose-6-phosphate from UDP-glucose and glucose-6-phosphate, and this intermediate is subsequently converted to trehalose by trehalose-6-phosphate phosphatase (TPP) [63,64,65]. This metabolic pathway is particularly active during vitellogenesis, as it fuels the accumulation of lipids and proteins in developing oocytes [66]—a role supported by studies in *Helicoverpa armigera*, where TPS/TPP transcription is upregulated during the pre-pupal and pupal stages to meet reproductive energy demands [67]. Vitellogenesis, another core process for insect reproduction, relies on the synthesis, transport, and deposition of Vg. Specifically, Vg is synthesized in the fat body, transported to the ovary via hemolymph, and taken up by mature oocytes through mediation by VgR; once inside oocytes, Vg precipitates and accumulates to form vitellin, an essential nutrient reserve for embryonic development [68,69,70]. In the present study, we observed that on day 3 post-injection of dsTPSs, the expression levels of both *Vg* and *VgR* in *N. lugens* were significantly downregulated compared to the control group (Figure 3E,F)—a result that links *TPS* silencing to impaired vitellogenesis. To further contextualize this finding, we referenced studies on the AKH signaling pathway, which interacts with trehalose metabolism to regulate insect reproduction. Previous work has shown that silencing the AKH receptor gene (*AKHR*) reduces *Vg* expression in the fat body and inhibits Vg deposition in oocytes, while concurrently decreasing trehalose levels in both fat bodies and hemolymph [71]; this reduction in hemolymph trehalose ultimately leads to decreased egg-laying and fecundity in *N. lugens* [72]. These results collectively suggest that inhibiting trehalose metabolism (either via *TPS* silencing or *AKHR* knockdown) curtails Vg deposition. In our study, *AKH* expression did not show a statistically significant difference between the dsTPSs group and the control (Figure 6D). This trend, combined with our direct expression of *Vg*/*VgR* downregulation (Figure 3E,F), may indicate a potential crosstalk between TPS-mediated trehalose metabolism and AKH signaling in regulating *N. lugens* reproduction. Additionally, relevant studies have shown that trehalase inhibitors block trehalase activity in eggs, preventing oocytes from absorbing required vitellogenin and thus reducing ovarian vitellogenin accumulation; this aligns with our finding that disrupting trehalose synthesis (via *TPS* silencing) impairs *Vg*/*VgR* expression, reinforcing the link between trehalose metabolism and vitellogenesis. This ultimately lowers egg hatch rates and curtails the reproductive capacity of *Spodoptera frugiperda* [73]. RNAi-mediated knockdown of G-protein coupled receptor genes (*AAEL003647* and *AAEL019988*) in *Aedes aegypti* reduced egg production by approximately 30% and induced abnormal ovarian development [74]. Vg synthesis occurs not only in the ovaries but also in the fat body. Notably, the TAG content in the ovaries was significantly decreased (Figure 6B). The insufficient energy supply derived from TAG in the ovaries directly inhibited either Vg synthesis or receptor-mediated endocytosis of Vg, ultimately leading to impaired yolk deposition [75]. In this study, injection of dsTPSs in *N. lugens* led to a reduction in egg-laying numbers and a significantly lower hatch rate compared to the control group (Figure 3B,C). This was accompanied by delayed ovarian development (Figure 2A) and egg abnormalities such as eye-spot inversion and eye-spot absence (Figure 3D). It is speculated that the silencing of the *TPS* gene leads to a decrease in *Vg* expression, which prevents oocytes from accumulating sufficient yolk. This causes oocyte development to arrest or degenerate, thereby resulting in reduced egg volume, thinner eggshells, decreased egg-laying quantity, and lower hatching rate. This also indicates that the inhibitory effect of the *TPS* gene on the reproduction of *N. lugens* is exerted by acting on egg grains.

The synthesis of Vg requires energy. Insufficient trehalose content leads to energy deficiency, which inhibits Vg synthesis. Meanwhile, *Vg* expression is also affected by a variety of signaling pathways. JH and 20E are two of the most critical insect hormones, orchestrating the entire reproductive process through intricate synergistic or antagonistic interactions [70,76,77]. JH activates the transcription of the *Vg* gene in adipocytes via the receptor Met, thereby initiating and sustaining vitellogenesis and regulating ovarian development during insect reproduction. Meanwhile, 20E not only governs insect molting but also plays a pivotal role in oocyte maturation and oviposition [78,79]. In this study, injection of dsTPSs in *N. lugens* significantly reduced ovarian area on days 3 and 5 post-injection (Figure 2A), increased the number of Grade I and II ovaries (Figure 2B), and decreased the number of mature eggs (Figure 2C). Concurrently, expression levels of *JHAMT* and *Met* were downregulated (Figure 4A,B), while key 20E signaling pathway receptors—*USP* and *ECR*—were also significantly reduced (Figure 4C,D). These results suggest that ovarian *TPS* expression is tightly regulated by JH: *TPS* silencing disrupts the intricate coordination and regulation between JH and 20E, arresting vitellogenesis and oocyte maturation and delaying ovarian development in female *N. lugens*.

Ovarian development is also influenced by nutrient signaling pathways. *N. lugens* has two insulin receptor genes, *InR1* and *InR2* [80]. RNAi results showed that *InR1* and *InR2* expression significantly decreased on day 3 post-treatment (Figure 5A,B). InR1 typically activates the PI3K/Akt pathway, while InR2 acts as a negative regulator of this pathway [80,81]. Studies have demonstrated that amino acids activate S6K via the TOR pathway, enhancing the translation efficiency of *Vg* and driving ovarian cell proliferation. Given the significant reduction in *TOR* expression observed in this study (Figure 5C), a corresponding decrease in S6K activity would be expected. Although *S6K* levels did not significantly change in this experiment (Figure 5D), a downward trend was noted. Silencing the *TPS* gene results in insufficient *TOR* expression, which in turn reduces Vg content, aligning with the observed trends in this study. The interplay between the insulin-TOR signaling pathways modulates lipid metabolism, thereby influencing genetic regulatory mechanisms [82]. After knocking down the expression of *PvVg* in *Polyrhachis vicina Roger*, the expression levels of genes related to other pathways all decreased significantly. *PvVg* regulates *PvERR* expression by crosstalk with the JH and IIS-TOR signaling pathways [83]. This is consistent with the findings of this study, and the decreased expression of the relevant genes is a phenomenon caused by specific regulation. Our study found that TAG content in the fat body remained unchanged, while the TAG content in the ovaries decreased significantly (Figure 6A,B). We hypothesize that when trehalose is insufficient, the energy supply in the fat body is deficient, which directly inhibits the expression or activity of lipolytic enzymes. This prevents TAG from being effectively decomposed and released from the fat body, thus keeping the TAG content in the fat body unchanged. The lack of energy provided by TAG in the ovaries directly inhibits the synthesis of Vg or the receptor-mediated endocytosis process, resulting in insufficient yolk deposition.

## 5. Conclusions

Collectively, our results demonstrate that trehalose-6-phosphate synthase (*TPS1*, *TPS2*, *TPS3*) acts as a central node integrating energy metabolism, hormonal signaling, and nutrient sensing to regulate *N. lugens* reproduction. Targeting *TPS* via RNAi-based strategies can effectively suppress *N. lugens*’s reproductive capacity without relying on traditional insecticides, thereby providing a novel, eco-friendly molecular target for *N. lugens* management. This not only offers a promising approach to mitigate *N. lugens*-induced rice yield losses but also aligns with the goal of sustainable agriculture by reducing environmental risks associated with chemical pesticides.

## Figures and Tables

**Figure 1 insects-16-01195-f001:**
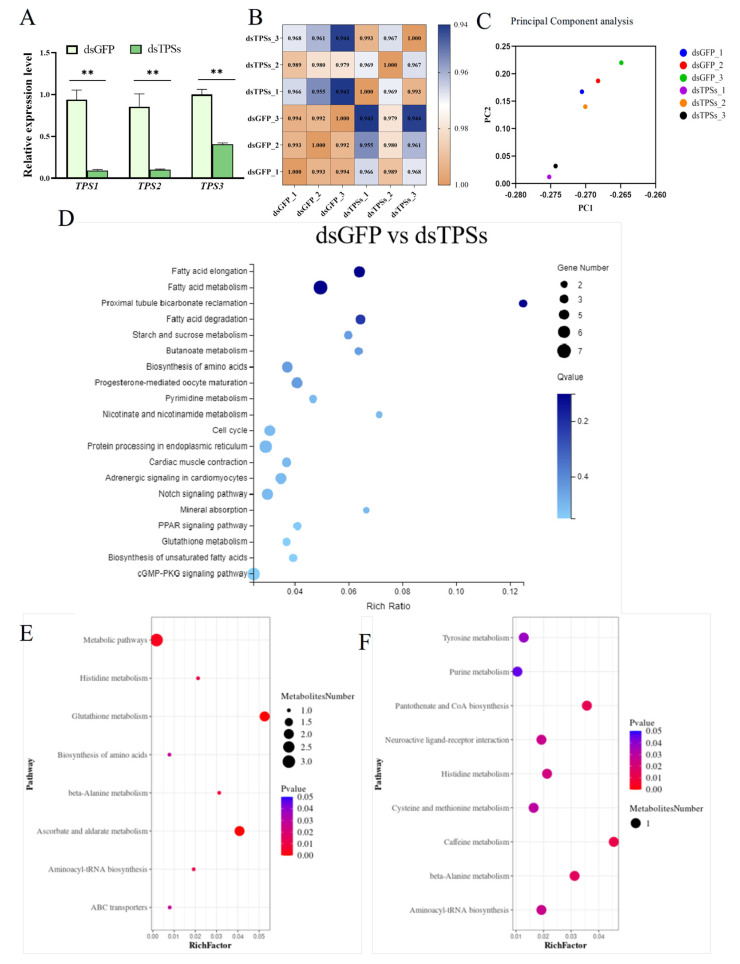
The relative expression levels of *TPS1*, *TPS2*, and *TPS3* in *N. lugens* on the third day after dsTPSs injection (**A**). Correlation analysis and principal component analysis among the transcriptome samples (**B**,**C**). KEGG metabolic pathway analysis of DEGs (**D**). KEGG pathway enrichment analysis of differential metabolites and analysis of crucial metabolites (**E**,**F**) ((**E**,**F**) represent the KEGG enrichment pathway between dsGFP and dsTPSs groups in the negative ion and positive ion mode). For the independent sample *t*-test, “**” indicated an extremely significant difference when *p* < 0.01.

**Figure 2 insects-16-01195-f002:**
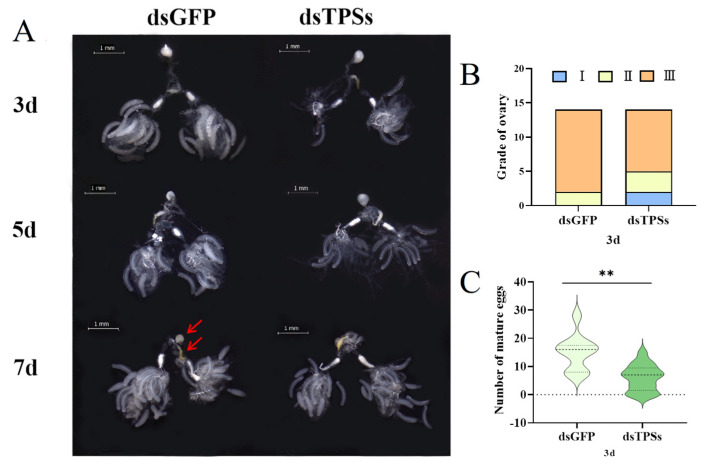
The ovarian development of *N. lugens* on the third, fifth, and seventh days after dsTPSs injection, magnifications of 16× were used (Scale bar = 1 mm). The white part indicated by the arrow is the bursa copulatrix, and the yellow part indicated by the arrow is the spermatheca (**A**), as well as the ovarian grade and the number of mature oocytes on the third day, N = 15 (**B**,**C**). For the independent sample *t*-test, “**” indicated an extremely significant difference when *p* < 0.01.

**Figure 3 insects-16-01195-f003:**
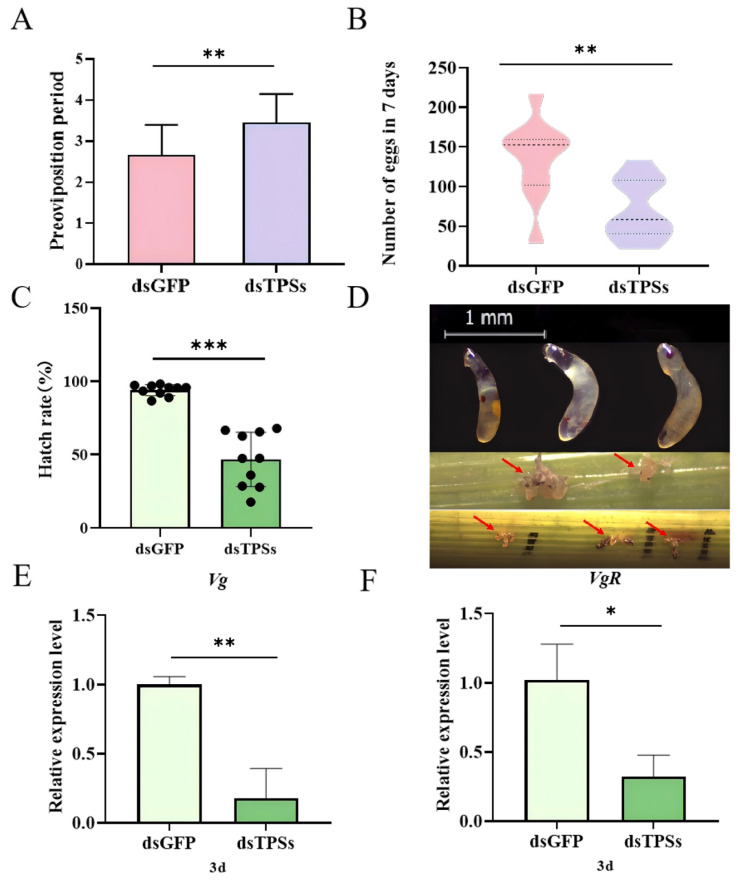
Preoviposition and total number of eggs in 7 days of *N. lugens* after dsTPSs injection (**A**,**B**). Hatching rate (**C**) and non-hatching phenotypes of *N. lugens* offspring, magnifications of 32× were used (The arrow indicates the unhatched egg grains in the rice) (**D**) and the relative expression of *Vg* and *VgR* (**E**,**F**) after dsTPSs injection. For the independent sample *t*-test, “*” denoted a significant difference when *p* < 0.05, “**” indicated an extremely significant difference when *p* < 0.01, “***”represented an even more extremely significant difference when *p* < 0.001.

**Figure 4 insects-16-01195-f004:**
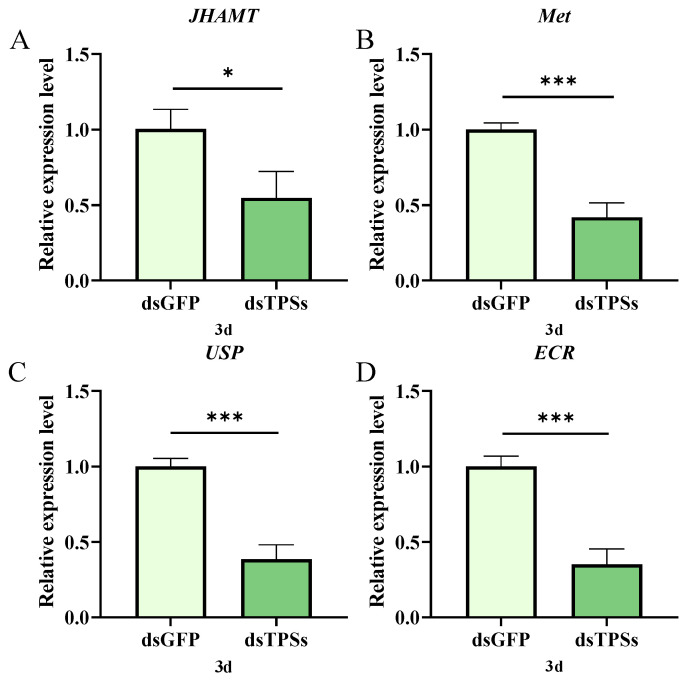
The relative expression levels of the juvenile hormone gene *JHAMT* and its receptor gene *Met* (**A**,**B**), the Ecdysone receptor gene *ECR* and the supervalve protein gene *USP* in *N. lugens* on the third day after dsTPS injection (**C**,**D**). For the independent sample *t*-test, “*” denoted a significant difference when *p* < 0.05, “***” represented an even more extremely significant difference when *p* < 0.001.

**Figure 5 insects-16-01195-f005:**
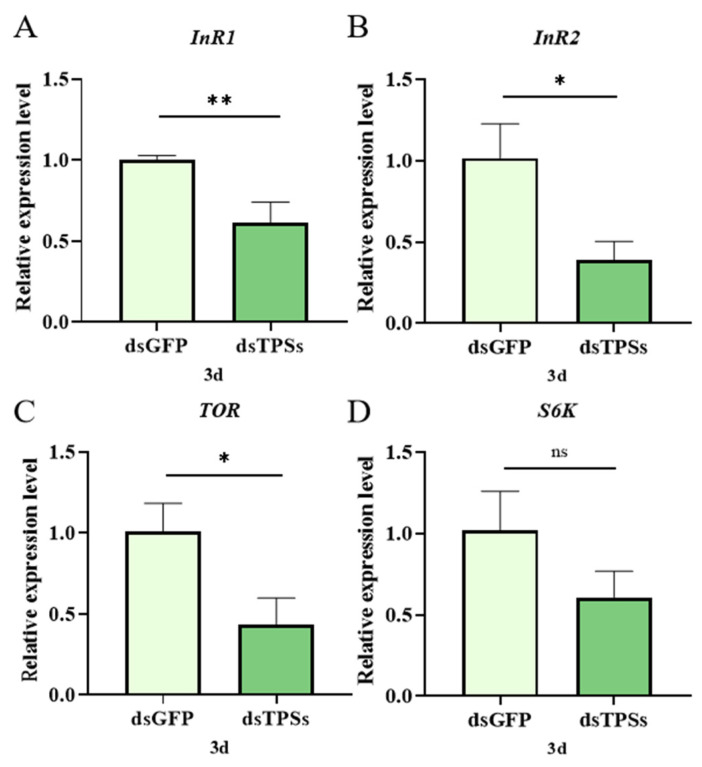
The relative expression of insulin-like peptide signaling pathway-related genes (*InR1* and *InR2*) (**A**,**B**) and target genes of rapamycin signaling pathway (*TOR* and *S6K*) (**C**,**D**) in *N. lugens* on the third day after dsTPSs injection. For the independent sample *t*-test, “*” denoted a significant difference when *p* < 0.05, “**” indicated an extremely significant difference when *p* < 0.01, and “ns” indicated no significant difference.

**Figure 6 insects-16-01195-f006:**
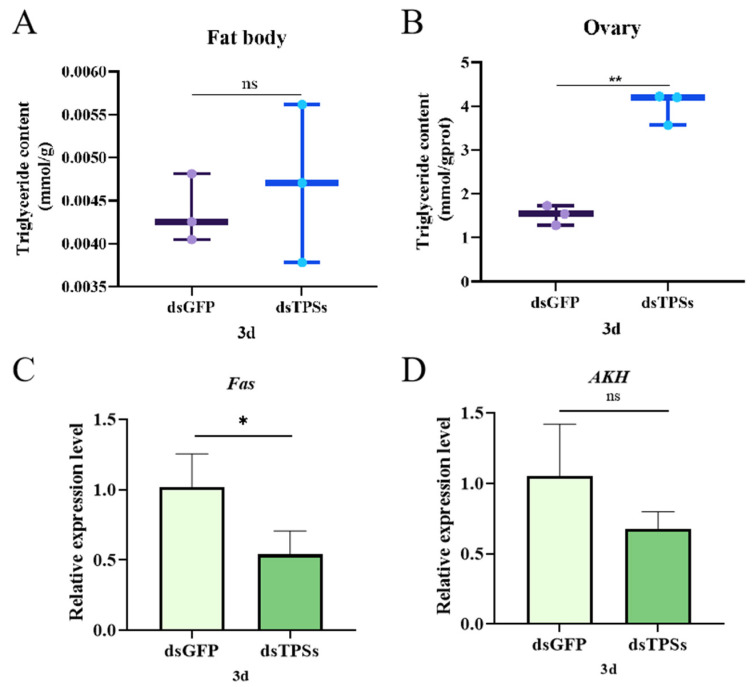
The relative expression levels of triglycerides in the adiposomes and ovaries of *N. lugens* on the third day after the injection of dsTPSs (**A**,**B**), as well as the relative expression of the fatty acid synthase gene *Fas* (**C**) and the lipid-activating hormone gene *AKH* in *N. lugens* (**D**). For the independent sample *t*-test, “*” denoted a significant difference when *p* < 0.05, “**” indicated an extremely significant difference when *p* < 0.01, and “ns” indicated no significant difference.

**Table 1 insects-16-01195-t001:** Primers for dsRNA synthesis.

Primer Name	Primer Sequence (5′-3′)
dsNlTPS1-F	ACCAGGAGTTGAAGGAGGAG
dsNlTPS1-R	CGATACCCGTGGGACTAG
dsNlTPS2-F	CACCAAAGGTCTAAGGCACA
dsNlTPS2-R	AGGGATGCTCTAGTTGCTAC
dsNlTPS3-F	GAGTCTGACCTGATAGCCTTTA
dsNlTPS3-R	TAGCCTCAGGTAAATCAACA
dsNlGFP-F	AAGGGCGAGGAGCTGTTCACCG
dsNlGFP-R	CAGCAGGACCATGTGATCGCGC

**Table 2 insects-16-01195-t002:** Primers for qRT-PCR.

Primer Name	Forward Primer (5′-3′)	Reverse Primer (5′-3′)
QActin	TGGACTTCGAGCAGGAAATGG	ACGTCGCACTTCATGATCGAG
QNlTPS1	AAGACTGAGGCGAATGGT	AAGGTGGAAATGGAATGTG
QNlTPS2	AGAGTGGACCGCAACAACA	TCAACGCCGAGAATGACTT
QNlTPS3	GTGATGCGTCGGTGGCTAT	CCGAACACGGTCCGCATA
QNlVg	CACTGCCCGTGCTGTGCTCTA	TGACTTCCTTGCTTTGCTCCC
QNlVgR	AGGCAGCCACACAGATAACCGC	AGCCGCTCGCTCCAGAACATT
QNlJHAMT	GAACCTGCAGGCCAAACACA	ACCACTCGGTTGGGCTGAAT
QNlMet	AGTGGCAGCGAGCGATGATT	TGAGGCGCAGCAAAAAGGAG
QNlUSP	GGTGGAGCTGCTGAGGGAGA	AGCACTTGAGGCCGATGGAG
QNlEcR	CGAAGCCTGGAAGGTGGAGA	GGCAAAGATTGGCGACGATT
QNlInR1	GAGTGCAACCCGGAGTATGT	TCTTGACGGCACACTTCTTG
QNlInR2	CTCTTGCCGAACAGCCTTAC	GGGTCGTTTAGTGGGTCTGA
QNlTOR	GGCTACAGGGATGTCAAA	GAGATAGATTCAAACGGAAAG
QNlS6K	AATCGGACGACTTGGAGACAGT	CAGTTTGGAAAGCGTACATCAGG
QNlAKH	CCCTTCTGATGGCAGTCCTTTG	ATGGATGCCTTGCAGCCTTCT
QNlFas	CGGAGACTCTGCCCTAA	CAGCGACTAATCCAACATC

## Data Availability

The datasets generated and analysed during the current study are available from the corresponding author on reasonable request.

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
