# Peer review of "Effects of Trehalose-6-Phosphate Synthase on the Reproduction and Development of *Nilaparvata lugens* and Its Molecular Mechanism"

_insects, 2025, doi:10.3390/insects16121195_

Round 1

Reviewer 1 Report

Comments and Suggestions for Authors

The paper is an interesting addition to the literature of N.lugens biocontrol. The targeting of trehalose synthase genes for control of N.lugens by reducing the fecundity is an interesting avenue of exploration. While it is clear to me that the work performed here was done very well and adds value to our understanding of N.lugens molecular biology I am not sure if the proposed gene targets will be valuable additions for proper biocontrol. The silenced genes do not cause mortality and while there is a significant reduction in both egg laying and hatch rate it is likely that this needs to be used in conjunction with other control methods. The discussion should explicitly address this. There are some minor issues that also need to be addressed:

  1. Line 44: Using the common name of Oryza sativa will improve the readability for the general audience particularly when in other places common names are used,
  2. Line 47: Please introduce the full name of BPH
  3. Line 94: Please add an appropriate citation
  4. Major comment: Connection to TPS and oviposition is not made clear in the introduction
  5. Line 125: The line is awkward, the method provides the result here it implies inversion
  6. Line 172: Please provide more details about the company sequencing what type of sequencing and with who
  7. Line 194: What commercial kit please provide more details
  8. Line 225: Kegg analysis is not mentioned at all in method section
  9. Categories of 1D is unreadable
  10. Same problem with 1E
  11. Ovarian grading is not cited or explained, please add this

Reviewer 2 Report

Comments and Suggestions for Authors

The authors present a well-structured and interesting workflow. However, the manuscript would benefit from a clearer and more concise presentation of the results and their broader implications. In addition, the manuscript requires careful revision of expressions and sentence structures. I recommend a thorough revision to address these issues.

Below are several specific comments and suggestions:

Lines 35–37: The expression “The synthesized dsTPSs silences the TPS gene” is not appropriate. The authors did not synthesize the gene itself; rather, they synthesized double-stranded RNA to suppress TPS mRNA. Please revise this sentence accordingly.

Introduction: It should be clarified when the authors refer to a specific insect species or if the results are generalized across insects. In several passages of the section, this is ambiguous and may confuse readers.

Materials and Methods: This section would benefit from greater specificity and more detailed methodological descriptions. Some concepts should be addressed or clarified. For example:

Section 2.2 (Line 126): TRIzol is a commercial reagent, not a method. Why the reference? Please review.

Section 2.3 (Line 134): Primers are used to amplify specific genes from a DNA template, followed by in vitro transcription to synthesize dsRNA. Please clarify this in the text. Also, Lines 137–140: The T7 Ribo kit should be cited with manufacturer information. If you mention some steps of the procedural, please give the context and details. If not, the mention need be removed –“Finally, an appropriate amount of DEPC water was added to dissolve the precipitate and stored at -80°C for future use”.

Section 2.4: Please indicate the RNA concentration/s and report the number of females (N) per treatment.

Section 2.5: Please standardize the terminology for quantitative real-time PCR. Indicate whether the reference gene was selected based on previous studies, or was selected among several candidates evaluated in this work. Primer efficiencies should be specified, and this information could be included in Table 2.

Section 2.6: The bioinformatic pipeline is not described. Please include details.

It is also unclear if the in-silico experiment was performed using the same sample of the qRT-PCR, please clarify that and justify. Report RNA concentrations, number of ovaries used, and sequencing service and platform. Also mention metrics and tools employed. The PCA and enrichment analysis need to be mentioned in this section.

Section 2.7: Please describe the criteria used to classify ovarian grades and report the number of females analyzed per treatment. Clarify how the pre-oviposition interval was recorded. Given that in phytophagous insects this parameter can vary depending on diet ingested, it would be important to explain how this variability was controlled.

Section 2.8 (Line 194): Please clarify “commercial kit.”

Results:

Section 3.1: The transcriptomic and metabolomic results lack explanation. It would be useful to include these data, possibly as supplementary tables, and indicate where the datasets are available (e.g., repository accession numbers). The description of the correlation, PCA, and enrichment analyses should also be included in the Materials and Methods section. Figure 1 should be improved in resolution and properly labeled axes.

Section 3.2:

Line 248: The expression “ovarian development was insufficient” and “development appeared normal.” are unclear, and should be revised for scientific accuracy.

The classification criteria for ovarian grades should be described.

In Figure 2C, the text refers to “mature oocytes,” whereas the figure mentions “mature eggs”—please ensure consistency.

Figure 2: Please indicate the number of females (N) per treatment in the legend, and clearly label the axes. Improve Figure resolution. Figure 2A should be enlarged to make structural details visible; with anterior and posterior orientations indicated. Please also include a scale bar.

Section 3.3:

Line 267: It is unclear whether the data on oviposition at day seven correspond to the same time point used to evaluate oocyte development in Section 3.2. This should be clarified to maintain consistency.

The embryonic results are not clearly presented, and the images do not provide conclusive evidence. Control images, scale bars, and a description of what the red arrows indicate should be included. Additionally, there is a distinction between embryo phenotype and egg phenotype, which the authors should take into account. Figure legends should consistently include information about the axes and experimental details.

Discussion:

The Discussion section would benefit from deeper interpretation and a more critical perspective. Currently, it mainly reiterates results rather than discussing their implications or comparing them with existing literature.

The conclusion that TPS regulate various pathways could be revised based on the fact that the results in this study show that all validated genes decrease expression. This could not be a consequence of the physiological state leading to lower overall transcription rather than a pivotal role in regulation. How would you justify this?

Additionally, the text mentions “egg abnormalities,” but these were neither detailed nor illustrated in the Results section. Moreover, the authors refer to defects in the chorion or embryo, without specifying or defining them. It may be relevant to discuss whether TPSs play maternal and/or embryonic roles, as suggested by the observed effects on oogenesis and embryogenesis.

Reviewer 3 Report

Comments and Suggestions for Authors

Summary.

In this manuscript, Han et al. investigate the effects of disrupting the expression of trehalose-6-phosphate synthase (TPS) genes on ovarian development. The authors used a transcriptomic approach to elucidate the molecular effects of TPS gene silencing and demonstrated that injection of dsRNA targeting three TPS genes significantly reduced ovarian development. The introduction is well written, and the results are compelling. However, several minor issues should be addressed to improve readability and clarify certain aspects of the results section. Additionally, the discussion should acknowledge some limitations of the study.

Minor comments.

Lines 219–220: The samples used for sequencing were ovaries, correct? I recommend briefly stating this in the text immediately following the citation of Figure 1A. This addition will improve readability. Currently, the section transitions directly into transcriptomic results without context, requiring the reader to refer back to the Methods section to confirm the sample source.

Figure 1. The figure appears somewhat blurry. Since the results are very interesting, I recommend improving the image quality to ensure that all details are clear and easy to interpret.

Lines 249–252: Please clarify in the material and methods section how Grade II and Grade III are defined. Are these classifications based on the number of mature oocytes, ovarian size, or other morphological criteria? If this grading system has been previously established, please include a reference in these lines to the source that originally defined or described these criteria in materials and methods.

Figure 3. Figure legend: Please indicate the meaning of the red arrows. Also, clarify whether the background black images represent close-up or zoomed in views, and specify which panel or area of the figure they correspond to.

Discussion section: The results are interesting, however, it is important to address the fact that three genes were targeted simultaneously. This approach makes it difficult to discern whether one gene contributes more significantly to the observed effects than the others. This limitation should be discussed. It would also strengthen the discussion to mention whether multiple TPS genes have been reported in other insects and, if so, how their functions compare. Furthermore, since these genes are expressed in multiple tissues, it would be valuable to comment on whether their expression levels differ in the ovaries, particularly if one gene shows higher expression, which could suggest a more prominent role in ovarian function.

Line 436: Since three genes are targeted by RNAi simultaneously, I recommend referring explicitly to all three in this line, i.e., “our results demonstrate that trehalose-6-phosphate synthase genes

Line 14. Define TPS abbreviation here.

Round 2

Reviewer 1 Report

Comments and Suggestions for Authors

While the scientific issues flagged in first submission is solved there still remains grammatical errors, specifically tense errors. Please give it one more round of screening. Also Primer Premier 5 and SOAPnuke software needs to be cited

Reviewer 2 Report

Comments and Suggestions for Authors

The authors have addressed several of the specific points raised in the review; however, some of the broader aspects that were suggested for a more comprehensive revision still require attention. In particular, the improvements regarding overall fluency and grammar are not yet fully reflected, and some non-academic expressions remain in the text. To assist the authors, I provide below specific suggestions and examples that may help guide a more effective revision, along with a recommendation to review the manuscript as a whole to ensure consistency and clarity.

General observations and stylistic notes

Gene names should be italicized, while protein names remain in regular font. Abbreviations should be introduced upon first mention and used consistently thereafter.

It would be helpful to replace colloquial wording with more formal scientific expressions throughout the text.

In several sections, sentences could be better connected to improve the logical flow of ideas. Adding linking phrases (e.g., furthermore, in addition, in contrast) would help guide the reader.

For methodological sections, please consider including additional details such as equipment models, sequencing depth, primer efficiencies, and statistical tests, which will improve reproducibility and transparency.

Line 25: Replace “strong” with “high”; consider rephrasing “outstanding drug by remarkable insecticide”.

Line 36: Replace “choreograph the entire reproductive program” with “orchestrate reproduction”.

Line 37: When dsTPSs (or any dsRNA) is first introduced, write the full name before using the abbreviation or acronym.

Line 42: Keywords may differ from words used in the title; choose keywords that maximize discoverability and reflect the manuscript focus.

Line 46: Replace “(rice plant)” with “rice”.

Lines 46–48: Simplify “(rice plant)” to “rice”; “yet its production” could be “However, rice…”; and “spectrum” might be replaced by “range” or “variety”.

Line 51: Introduce WBPH with a proper citation when first used (e.g., full species name).

Lines 52–61: These sentences require linking to show how ideas connect. Add transitional phrases (e.g., “Furthermore,” “In addition,” “Consequently,” “However,” etc.) to build cohesion.

Lines 65–66: Replace “dissecting gene function and is now…” with “functional genomics and a transformative platform for…” Remove “next generation”

Line 68: “for delivery”

Lines 69–70: Connect this sentence to the previous idea.

Line 72: Remove Harris; change “. Silencing” to “, while suppression of”.

Line 73: Replace “Silencing” with “Similarly, silencing”.

Line 74: Species name to “S. frucifera” and replace “yields a lethal phenotype” with “leads to lethality”.

Lines 74–76: Link this statement to the preceding sentence.

Lines 80–83: Clarify whether TPS paralogs are tissue-specific or ambiguous across tissues. Replace “three distinct TPS” with “three TPS”.

Lines 86–92: These sentences are presented as a list of facts with no connectors. If the aim is to highlight multiple functions of trehalose, add transition phrases to compare or contrast functions across contexts.

Line 93: Specify which insect(s) are being referenced.

Line 99: Replace “In choice” with “Behavioral assays”; replace “both” with “show that”; correct “its congener S. frucifera” to “S. frucifera”.

Line 101: Replace “whereas” with “while”; change “will continue” to “continues”.

Lines 101–110: Revise for clarity.

Line 111: Remove the sentence.

Lines 114–115: Clarify the manuscript objective: is it to identify a target gene or to propose a control strategy using that gene? Make the objective congruent with the Results and Conclusions sections.

Line 160: Section heading should read: “2.5 Quantitative real-time polymerase chain reaction (qRT-PCR)”. Report primer efficiencies for each primer pair.

Line 161: Use consistent terminology: replace “qPCR” with “qRT-PCR”.

Line 173: State which stereomicroscope model was used

Line 175: Provide specifications for sequencing libraries, read depth, and sequencing platform. Provide equivalent details for metabolomics.

Lines 181–188: Include citation and version for each program and software used. For metabolomics, include methodological descriptions.

Line 226–227: Report whether normality tests were performed; which ones?

Line 236: In subsection “3.1 RNAi efficacy and transcriptomic-metabolomic analysis”, report sequencing metrics (total reads per sample, mapped reads, coverage, quality scores) and metabolomics run metrics.

Line 246–247: This statement reads as methodological detail not results.

Lines 251–254: Clarify whether the three TPS genes’ expression was also evaluated using in-silico datasets.

Line 257: Figure 1 D, E and F legend labels are illegible — increase font size and ensure readability.

Line 278: Figure 2 A — images of ovarioles are too small to appreciate differences between treatments; increase image size or provide inset close-ups. Include scale bars in the images and indicate scale (e.g., 200 µm); do not only report magnification — use scale bars in the figure and mention scale in the legend.

Lines 289–290: The phrasing of this line reads as Materials & Methods; keep Results section focused on observed data.

Lines 292–294: The authors must clarify whether the observed lower hatching rate is a consequence of reduced oviposition (fewer eggs laid) or due to lower hatchability of the eggs produced. Provide raw counts (table of crude values) so readers can evaluate this distinction.

Lines 296–298: Relocate this results because the section 3.3 is focused on egg-level results.

Line 300: Figure 3D — include a scale bar (not only magnification).

Line 310: Avoid the word “suppressed” for gene expression results; it implies complete inhibition. Use “reduced” or “decreased relative to control”.

Lines 308–313: Results are repeated across sentences; consider consolidating and unifying statements to avoid redundancy.

Line 341–343: This statement is not a result. Rephrase.

The Discussion section contains some grammatical mistakes and would benefit from a careful revision for clarity and coherence. In particular, between lines 371–418, some statements are not fully supported by direct evidence from the presented results. Additionally, the description of the different metabolic pathways appears somewhat mixed, making interpretation challenging. The authors are encouraged to revise this section for clarity and logical organization.

Moreover, the observed reduction in mRNAs involved in energy metabolism after TPS gene silencing may reflect a general metabolic stress response rather than a direct inhibition of vitellogenesis. Considering and discussing this alternative explanation could strengthen the interpretation. Finally, please clarify whether vitellogenin (Vg) production in N. lugens occurs exclusively at the ovarian level, and provide supporting references where appropriate.

In reference to Comment 14 from first review: “Comments 14: The Discussion section would benefit from deeper interpretation and a more critical perspective. Currently, it mainly reiterates results rather than discussing their implications or comparing them with existing literature.

The conclusion that TPS regulate various pathways could be revised based on the fact that the results in this study show that all validated genes decrease expression. This could not be a consequence of the physiological state leading to lower overall transcription rather than a pivotal role in regulation. How would you justify this?

Response 14: Thank you for your question, we have made some supplements in the manuscript. As an important energy substance in N. lugens, trehalose, when its synthesis is silenced, will directly affect the JH signaling pathway and the target of TOR signaling pathway. The precise downregulation of genes in these two pathways is consistent with the specific regulation patterns reported in other studies.”

Given that all validated genes showed decreased expression, it is still unclear whether this pattern reflects a specific regulatory effect of TPS silencing or rather a secondary consequence of a general metabolic slowdown or energy stress caused by reduced trehalose synthesis.
